# Reduction Expansion Synthesis of Sintered Metal

**DOI:** 10.3390/ma12182890

**Published:** 2019-09-06

**Authors:** Zachary Daniels, Wilson Rydalch, Troy Y. Ansell, Claudia C. Luhrs, Jonathan Phillips

**Affiliations:** Mechanical and Aerospace Engineering Department, Naval Postgraduate School, Monterey, CA 93943, USA (Z.D.) (W.R.) (T.Y.A.) (C.C.L.)

**Keywords:** cold-sintering, additive manufacturing

## Abstract

Described herein is a novel method, Reduction Expansion Synthesis-Sintered Metal (RES-SM), to create a sintered metal body of a designed shape at ambient pressure, hundreds of degrees below the metal melting temperature. The precursor to the metal part is a mixture of metal oxide particles and activated metal particles, and in this study specifically nickel oxide and activated nickel metal particles. It is postulated that the metal oxide component is reduced via exposure to chemical radical species produced via thermal decomposition of urea or other organic compounds. In the study performed, the highest temperature required was 950 °C, the longest duration of high temperature treatment was 1200 s, and in all cases, the atmosphere was inert gas at ambient pressure. As discovered using scanning electron microscopy (SEM), energy dispersive spectroscopy (EDS) and x-ray diffraction (XRD), the metal that forms via the RES process presents necks of completely reduced metal between existing metal particles. The ‘as produced’ parts are similar in properties to ‘brown’ metal parts created using more standard methods and require ‘post processing’ to full densify. Parts treated by hot isostatic pressing show fully self-supporting, robust structures, with hardness values like others reported in literature for traditional fabrication methods. This novel method uses affordable and environmentally friendly precursors to join metallic parts at moderate temperatures, produces fully reduced metals in a very short time and has potential to make many parts simultaneously in a standard laboratory furnace.

## 1. Introduction

This work was designed to test a simple hypothesis. The Reduction Expansion Synthesis [1,2,3,4,5,6,7,8] concept can be employed in a novel variation, Reduction Expansion Synthesis-Sintered Metal (RES-SM), to create, quickly, fully reduced ‘brown’ self-supporting, metal only objects of designed shape at ambient pressure, and temperatures hundreds of degrees below the melting temperature of the metal. As described below, this postulate proved true.

In addition to testing and demonstrating the validity of the main hypothesis two additional significant phenomena were recorded. First, typical for a brown metal part, treatment using hot isostatic pressing (HIP), again at hundreds of degrees below the melting temperature of the metals employed, transformed the brown bodies created with RES-SM to near full density metal parts with properties equivalent to bulk metals of similar shape created by casting from liquid metal. Second, using multiple RES treatments accelerates sintering. Specifically, hardness and density measurements showed that brown parts re-baked using the RES protocol densified and hardened faster than simply heating at the same temperature in an inert atmosphere.

The general RES concept in broad terms: Chemical radicals are released by thermal decomposition in an inert atmosphere of some simple chemicals such as urea and petroleum gel. The released radicals react with oxygen atoms in nearby solids to form volatile oxygen complexes, leaving reduced metal behind. An exemplary example: A physical mix of urea and metal oxide (e.g., iron) particles, heated to 950 °C in an inert atmosphere, creates fully reduced metal particles [2,3].

The production of macroscopic scale, solid metal parts, a simple modification of this general mechanistic hypothesis is advanced: mobile metal atoms will form when metal oxide particles in a compact of metal and metal oxide particles are exposed to a reducing atmosphere. In a process akin to classic Ostwald Ripening [9,10], these atoms, or atomic clusters, effectively radicals, will migrate to, then bond to, existing metal surfaces in the immediate vicinity to create larger metal bodies. A similar ‘metal radical’ model is used to explain the growth of large metal particles in the gas phase during catalytic etching [11], as well as enhanced rates of particle growth in supported catalysts under reaction conditions [12]. A test of this hypothesis: Using the RES-SM method a solid object can be generated from a mixture of metal oxide and metal particles, and not from metal particles alone. 

Observations in this study regarding brown body formation are consistent with the above model. Most important, larger bodies do form from a metal/metal oxide compact with a morphology consisting of the original metal particles joined by metal ‘necks’. That is, the ‘brown body’ consists solely of a network of metal particles linked by ‘necks’, and a homogenous distribution of void spaces of about the same size as the original particles. Moreover; the final metallic object has the same ‘shape’ as the original compact but is smaller in size. The compact shrinks in a uniform fashion such that relative dimensions remain nearly unchanged. Hot isostatic pressing creates a part with hardness equal that of metal made by standard casting. In contrast, using the RES-SM protocol a solid object does not form from metal particles only. 

## 2. Materials and Methods 

The process consists of these steps: (i) making an ‘activated’ metal/metal oxide particle compact, (ii) arranging the compact and the reductant appropriately in a furnace, and (iii) heating the compact and reductant for a short time in an inert atmosphere at ambient pressure at a temperature far below the metal melting temperature.

### 2.1. Activation 

Five mL of 30% Hydrogen Peroxide (30% in H_2_O, Millapore-Sigma, Darmstadt, Germany) are mixed with each gram of metal nickel powder (5 µm, 99.7% purity, Millapore-Sigma, Darmstadt, Germany) then heated on a hot plate to ~70 °C. The reaction was determined to be concluded (~300 s) when bubble formation stopped.

The activation clearly only modifies the surface of the metal particles. Neither XRD nor EDS performed in the SEM was able to detect any change in the particles. For example, only Ni metal and no oxide was detected before or after activation. It is postulated that the primary role of the activation is surface cleaning. That is, activation removes carbonaceous material commonly found on all surfaces, creating a clean surface for the RES-SM process.

### 2.2. Particle Compact

The starting material for the particle compact in all cases was a physical mixture of nickel oxide (Sigma Aldrich, 44 µm, 99% purity) and the activated nickel metal particles ground together briefly (ca. 5 min) using a mortar and pestle. Several weight ratios of the metal/metal oxide were tested. Also controls, that employed a different protocol, described in the Results section, were performed. None of the control protocols created Ni metal, self-supporting objects.

In one example a roughly ‘rectangular’ compact, 3 cm × 1 cm × 2 mm deep, was created by hand shaping a 1:1 weight mix of Ni:NiO. The shape was formed on top of a section of perforated Grafoil (4 cm × 2 cm) ‘support’ (Figure 1). In other cases a tensile specimen, with the dimensions ratios employed for the ASTM International standard flat ‘dog bone’, compact roughly 3 cm long was created by filling a mold produced using a standard desktop 3D printer (Ultimaker 3 Extended 3D printer, using 3 mm PLA filament, Ultimaker B.V., Utrecht, Netherlands), positioned on top of a perforated 0.3 mm thick Grafoil, a ‘paper consistency’ material composed (99.9%) of compressed Graphite flakes with a surface area of 22 m^2^/gm [13,14]. Before firing, the mold was removed, yet the shape was retained with no binding agent.

### 2.3. Firing 

Once the shape was formed the next step was ‘firing’. The standard process employed is as follows. (i) Arrange the urea (Sigma Aldrich, beads ~99% purity) in the bottom, and shaped metal precursor on a perforated Grafoil® (GTA Grade 0.3 mm thick, NeoGraf Solutions, Lakewood, OH, USA) top in an alumina boat as shown in Figure 1. (ii) Place the alumina boat in the center of a 50 cm × 2.5 cm diameter quartz tube. (iii) Flow UHP nitrogen (Praxair, Salinas, CA, USA) through the quartz tube at ~50 sccm for 10 minutes. (iv) Reduce nitrogen flow to ~10 sccm and place quartz tube inside a 30 cm long tube furnace v) Ramp heat the tube furnace (Lindburgh-Blue M 24” single zone) to 950 °C and hold for ~1200 seconds. (vi) Quickly remove the quartz tube, nitrogen still flowing, from the furnace. (vii) Continue to flow nitrogen, while allowing the quartz tube and finished sintered body to cool at ambient temperature (ca. 20 minutes). Open tube, remove and examine product. In some cases, as described below, the same sample was repeatedly re-fired using the above protocol.

In all cases 0.5 g urea was present beneath, and the NiO:urea ratio (except controls) was ~1:1. The constant urea load was selected as the maximum amount consistent with the glass system remaining sealed. Indeed, the decomposition process creates a pressure burst, but tests showed for 1 g urea, glass joints, at tube entrance and exit, were not moved. A smaller amount, 0.5 g, was felt to allow a good safety margin.

### 2.4. Controls

A number of controls, variations on the standard protocol, were performed. None of the control protocols created Ni metal, self-supporting objects. In all control cases only loose, unbound particles, or weakly bound small clusters (<0.5 cm in dimension) were generated. Control 1: Standard protocol, but only activated Ni metal particles were in the precursor. Control 2: Only NiO particles were in the precursor. Control 3: The precursor consisted of a 1/1 mixture of Ni:NiO particles, but no urea was present underneath. Control 4: The precursor consisted of a 1:1:1 intimate mixture of Ni:NiO:urea. There was no urea under the precursor. 

### 2.5. HIP 

In addition to the controls, in one case (more below) a dog-bone shaped sample was Hot Isostatically Pressure (HIP) treated at 2,040 atmospheres pressure in inert gas for 10 h at 1000 °C. 

### 2.6. Analysis

Physical and chemical analysis tools were used to characterize sample morphology, composition, and strength. Five different tools were employed. (i) X-ray powder diffraction (XRD) was performed on samples using a Rigaku Mini-flex 600 X-ray diffractometer (Rigaku Corporation, Tokyo, Japan) operated at 40 kV and 15 mA with a Cu metal target (1.54 Å Kα line). Diffraction data was collected in the 2θ range of 10° to 90° at 3–5°/minute with a step width of 0.02°. Diffraction data analysis and structural refinement were performed using Jade 9. (ii) Morphology of Ni specimens was investigated using a Zeiss Neon 40 scanning electron microscope (ZEISS International, Oberkochen, Germany) with a 30 μm aperture and an accelerating voltage of 20 kV. Images from the SEM were used to make a qualitative assessment of the level of sintering in samples. (iii) Chemical analysis was performed using an EDAX energy dispersive spectrometer (EDAX Inc., Mahwah, NJ, USA) attached to the SEM. (iv) The density of the products was determined using an Ohaus Density Determination Kit which employs a simple buoyancy method, combined with direct weight determination. In this work distilled water was used for the buoyancy determination. (v) Hardness was measured using a Struers DuraScan Vickers Micro-hardness tester using 0.1 kgf.

## 3. Results

Four different control cases were studied. In all cases the temperatures and times employed were the same as in the standard protocol, only the ‘chemistry’ of the standard process was modified. None of these control protocols produced a solid, self-supporting object. In only Control 4 (see Experimental) was NiO reduced to Ni; however, the final product was a loose bed of particles, not a solid object. In sum, these controls clearly demonstrated that to make a solid metallic object the precursor must contain both Ni and NiO, and urea must be present, but not mixed into the precursor. The samples created using the standard protocol described above are all listed in Table 1.

Samples containing both nickel oxide and nickel metal particles, with urea in the alumina boat beneath, generated solid, self-supporting bodies (Figure 2). The color change, observed in all cases, clearly suggest Ni oxide (green) has been reduced to metal (‘silver’), a conclusion consistent with XRD. It is apparent (Figure 2) that the rectangular hand shaped structures spreads, both horizontally and ‘vertically’, during processing. In contrast, less spreading and generally fewer imperfections are observed in the dog bone. This is attributed to the ability to apply greater pressure, and to apply it more evenly, to a precursor in a mold. Taken together, these preliminary efforts suggest protocols for even more perfect shapes can be developed.

Significant shrinkage occurs during firing. Careful measurements of ‘dog bones’ created with different Ni:NiO concentrations indicate that in all cases the final solid product had a dimension of 75+/−3% of the original particles-only body. Moreover; the shrinkage occurred uniformly in all dimensions. Note: This significant shrinkage is consistent with measured density (Table 2), as density was only measured after firing. 

It is also informative to compare the top and bottom sides of the self-supporting body formed during firing. As shown in Figure 3 the two are not the same. Exemplary is the rectangular body. The top is smooth, with some cracks and scratches and a small (ca. 1 mm) thin ‘layer’ extending beyond the ‘bulk’ structure at the edges, whereas the bottom additionally shows dimples which formed directly above the pin holes in the underlying Grafoil.

XRD (Figure 4) shows virtually 100% reduction for the 3:1 Ni:NiO precursor mixtures after a single firing, but not for the 1:1 Ni:NiO precursor mix. For the precursor mixes containing more than 50% NiO, XRD suggests ~95% reduction to metallic nickel. However; the NiO peaks disappeared after a second firing. The reduction of the NiO to Ni is consistent with the expectation of the RES process. That is, the radical species generated by the thermal decomposition of the urea interact with oxygen in the nickel oxide to form volatile oxides, such as CO_2_, leaving reduced metal atoms behind.

The SEM results are also consistent with the RES-SM hypothesis. As shown in Figure 5, the process creates a homogeneous morphology consisting of metal particles, ‘necks’ between metal particles and void spaces of a similar size scale. The existence of necks between particles is a classic feature of sintered metal particles [15,16,17]. As shown, metal particles are generally joined to at least two other particles and often many more. Void space is an inevitable outcome of the sintering of particles, as contrasted to nucleation and solidification from a liquid.

Chemical analysis of the product using EDS was employed to confirm the particle surfaces were fully reduced. EDS, a surface sensitive technique, compliments the bulk characterization of XRD. There is no evidence of any significant impurities in the particles. Neither oxygen nor carbon was discovered in significant quantities. Even for the once-baked 1:1 Ni:NiO sample (Figure 6), EDS showed miniscule carbon (~1.0 wt %) and oxygen (~0.1 wt %) content. Small amounts of Al and Si were also detected; however, the quantities of these contaminates were well below 0.1 wt % in all observed spots of both the once- and twice- baked 1:1 samples. Limited oxidation of metal surfaces is always anticipated. So, the existence of some oxygen in the surface is consistent with complete reduction during the baking steps, and some oxidation occurring naturally during subsequent air handling.

After HIP treatment, the particles have a complex, inhomogeneous structure, apparently a mix of fully densified nickel zones and large voids. As shown in Figure 7, collected in SEM after cutting a HIP’d sample, there is no void space between particles in some areas, but voids still exist (more below). Moreover, after the HIP treatment, the density is still of the order of 95% (Table 1), consistent with the existence of significant void inclusion.

To make a better comparison of the microstructure of the HIP treated sample with traditionally prepared cast samples, the HIP treated material was polished using standard metallographic procedures. When examined in the SEM, the sample does not have a uniform distribution of voids and particles. On the size scale shown, the morphology consists of zones with solid nickel and some micron scale voids (Figure 8).

In order to make a quantitative comparison of mechanical properties with cast nickel, the specimen’s Vickers hardness was measured using a Micro-hardness tester. Multiple measurements were conducted, resulting in an average hardness of HV 86 with a standard deviation of HV 4. Recent measurements of the Vickers hardness for cast, but unworked, nickel are all between HV 81 and 90 [18,19,20].

Finally, it was noted that each firing reduced the physical size of the dog bone, as recorded in Table 2. The rate of shrinkage and the rate of hardening both are accelerated by use of urea.

## 4. Discussion

### 4.1. Brown Body 

The key observations regarding the formation of a hard, self-supporting, virtually 100% nickel metal object from a mixture of NiO and Ni metal particles undergoing the RES-SM protocol described above are the following:A solid body only forms if both metal oxide and metal particles are present in the precursor. This finding strongly supports the hypothesized mechanism (Section 4.2) of RES-SM;A solid body only forms if urea is present, and the urea is not mixed with the metal precursors;At least 90% of the oxide in the particle compact is reduced to metal, and after ‘double firing’ only reduced metal is present in all cases;The process leads to the formation of metallic necks between the original metallic particles in the precursor. Careful dimensional measurement and mass determination indicate the brown body density is of the order of 85%;Significant shrinkage occurs during the formation of the solid ‘brown body’ from the particle compact. Linear dimensions of the final product were approximately 75% those of the original particle compact.

It is also important to note that the process included some fundamentally novel features. First, a ‘brown’ [21] body is formed from a mix of metal and metal oxide. In all other systems, including particle injection molding (PIM) and all M-AM technologies, the metal precursors are entirely reduced metal particles. Second, no ‘binder’ is required as per PIM or metal containing paste based AM techniques. This is particularly important as ‘debonding’, that is removal of the binder, generally an organic material, is the slowest, most complex step in these PIM processes. Thus, in the terminology of PIM, the body formed in RES-SM is a ‘brown’ body, that is it is equivalent to a de-bonded PIM generated body, a body consisting only of metal. Third, unlike M-AM, at no point is there a need to reach the metal melting point. Indeed, in this study, a solid nickel body formed at 900 °C, whereas the nickel melting temperature is 1455 °C.

### 4.2. Model

A simple chemical model of the process helps explain the basic observations outlined above. The chemistry, part of a new class of chemical processes known as Reduction Expansion Synthesis (RES), is based on this model: Radical species, created during the thermal decomposition of a ‘solid reductant’ material, diffuse to an area containing ‘reducible’ metal oxides. The radicals interact with oxygen in the metal oxides to form CO_2_, etc., leaving reduced, mobile, metal atoms.

This model is particularized to RES-SM in which urea was the solid reductant. Above ca. 300 °C urea begins to decompose, fully decomposing in stages (18,19) as outlined elsewhere. This thermal decomposition releases volatile radicals, including CO and NH_x_ groups. The radicals travel to the ‘nearby’ compact body composed of nickel and nickel oxide. The radicals react with the oxygen in the NiO, forming a volatile gas species such as CO_2_, NO_x_, etc. The nickel oxide is thus reduced. The reduced Ni metal atoms (or small cluster) created by this process bond to other nearby Ni metal surfaces. That is, the atoms/clusters migrate either along a solid surface, or through the gas phase, until it finds a suitable ‘template’, which is an existing Ni particle surface. This process is repeated until all the NiO is reduced. The described process, atom by atom addition, is essentially identical to the classic model of particle growth known as Ostwald Ripening [9,10]. The addition of Ni metal atoms to the existing metal particle surfaces creates ‘necks’ or bridges between adjacent Ni particles. Moreover; the significantly reduced dimensions of the self-supporting body relative to the particle compact indicates the process actual ‘pulls’ reduced metal particles together.

The design of the reactor required to create solid bodies is based on the above model and ‘understandings’ reached based on earlier work. For example, in the RES-SM process the solid reductant was not mixed with the metal precursors. The decomposition process was physically isolated from the particles to be sintered. This was done because an early RES process in which similar precursors and urea were physically mixed led to particle production, not solid body production. Specifically, submicron scale nickel metal particles were created from micron scale nickel hydroxide and nickel oxide particles physically mixed with urea and heated to ca. 950 °C for a few hundred seconds under flowing nitrogen gas [2,3,4]. Moreover; it was shown that alloy particles could be formed using this approach. Indeed, submicron sized Fe-Ni alloy particles were produced from physical mixtures of urea, nickel nitrate and iron nitrate. Those studies show that if urea is mixed with the precursors, the gas expansion associated with urea decomposition ‘blows’ particles apart, preventing particle sintering. Indeed, one of the control studies (Table 1), basically a repeat of the work done earlier [2,3,4], confirms that physical mixtures of metal precursors and urea only produce particles. Hence, in the present case, a method was devised to allow the urea decomposition products to interact with metal precursors yet prevent gas expansion from inhibiting sintering. Those studies were also valuable in the design of the present process as they validated the postulate that gas phase radicals produced via the thermal decomposition of urea, will react with oxygen in metal oxide particles to form volatile oxides, leaving metal behind.

Other earlier studies helped validate the general RES mechanism and contributed to the general understanding of the RES process, enabling good ‘engineering’ of the RES-SM process. Example: The RES process was employed to create graphene from graphene oxide [1]. Again, the process followed the same basic pattern: (i) Physically mix urea and the ‘parent oxide’, graphene oxide. (ii) Heat the mixture in an inert atmosphere at a temperature sufficient to thermally decompose the urea. The urea decomposition products reduce the parent oxide, leaving graphene. Another example: A mixture of activated carbon (high concentration of surface oxygen groups), urea and tin chloride were heated to about 900 °C in an inert atmosphere. The resulting Sn/carbon electrode was shown to be much more stable as a battery anode than any prior Sn/C electrode [8]. That is, anodes made in this fashion lose less than 20% of capacity over about 200 cycles. In contrast, it has been repeatedly shown that Sn anodes produced using other methods fail after less than 10 cycles. It was postulated this resulted from the following process: (i) The urea decomposition products removed oxygen groups from the activated carbon, (ii) the dangling carbon atoms (‘surface radicals’) left by the removal of surface oxide groups [9,10] reacted with Sn atoms released by the decomposition of SnCl. This resulted in a unique, strong, direct bond between carbon and Sn metal. Activity, stability and TEM observation of very small particles that sintered very slowly, were all consistent with this hypothesis regarding RES generated Sn/carbon. In contrast, it is postulated that other Sn/carbon deposition processes create an oxygen linkage between the metal and carbon. There is no metal-carbon bond formed during standard Tin electrode generation. Another example of RES is the production of a metal coating, specifically chrome coating on iron [5,6]. Yet another example is a RES process to make a remarkably small, (ca.<10 atom) Pt particles on Mo_2_C, an electrical conductor. This material was found to function as an extraordinarily stable fuel cell electrode [7]. The RES-SM method is a straight forward extrapolation of general understanding garnered from these earlier efforts.

### 4.3. HIP Morphology 

The HIP process employed essentially consolidated the metal particles and the void spaces. Prior to HIP the body is a homogenous mixture of voids and necked metallic nickel particles, whereas after HIP the body is inhomogeneous. There are clearly areas where the Ni particles have sintered together, but there are also voids of a far larger volume than any voids found in the brown body. The HIP process led to the removal of the connected pore network and created high net density and hardness (away from voids) equivalent to that reported for cast nickel.

### 4.4. Accelerated Sintering 

At the temperature employed herein, 900 °C for RES-SM firing, compacts of pure nickel particles *(no oxide particles)*were shown in prior studies to sinter without a reducing agent at high pressure [22,23]. It is notable that quantitative comparison with earlier studies is not possible because all of them were conducted at pressures at least 1000× higher than that employed herein. Moreover, it is clear that high pressures (ca.>1000 atm), even at 600 °C, dramatically impacts densification [22]. Still this raises a question: After the first firing, necessary to reduce all the nickel oxide, is there any advantage to subsequent firing? Table 2 clearly suggests there is a case to be made for accelerated sintering using standard RES-SM firing. Indeed, the hardness and ‘shrinkage’ after 60 minutes at temperature in the RES-SM process is far better than that achieved in inert gas at the same temperature for 120 min. Possibly further study will reveal a path to full densification, without HIP, at ‘low’ temperature and ambient pressure.

### 4.5. Additive Manufacturing

RES-SM can be considered a form of metal additive manufacturing as no material is ever removed. Notably, it is related to another ‘hybrid AM’ technology based on creating molds with P-AM and adding metal. Specifically, parts are created by pouring liquid metal from low temperature melting metals (bismuth, lead, tin, cadmium, and indium) into plastic molds with a top operative temperature of ~180 °C [24]. The advantages of RES-SM are: (i) Low cost of the hardware, (ii) lower cost of the precursors (metal oxides are less expensive than metals), (iii) simplicity, (iv) scalability (e.g., many parts can be manufactured simultaneously in a single furnace), (v) rapidity as only 20 minutes are required for processing a part of any size, and (v) the wide pallet of metals (e.g., Ni, Fe, Co, Sn, Cr) and alloys that likely can be used to create solid objects with the technique.

Given RES-SM is a form of M-AM, could it potentially be developed into a competitive technology? At this stage of RES-SM development an answer is premature. Even the ultimate thickness of articles that can be made using this technique is unknown. Still it is reasonable to point out that that relative to one technique, M-AM by selective laser raster of a particle bed, first introduced thirty years ago, there may be advantages to RES-SM. To wit: Assuming equal mechanical properties can be achieved using either RES-SM or a commercial layer-by-layer laser/e-beam sintering, at present the former represents a far smaller (ca. 5%) capital investment, is far faster (~20 min., any size object), and potentially can be used to make large objects. The latter, presently, has the advantages of more optimized operation, better precision, hence more detail in the product, clearly can make objects many centimeters in all dimensions, and likely creates a superior product. Also, any M-AM process that requires high pressure and/or debonding (e.g., PIM), is likely slower and more costly than RES-SM. However, it is clear more M-AM technologies will be introduced in the near future, that all M-AM technologies are rapidly transforming/improving, so all comparisons and projections are fraught with uncertainty.

## 5. Conclusions

The data strongly support the hypothesis of RES-SM: A mixture of metal and metal oxide particles, exposed to the volatile species created by the thermal decomposition of urea, remove oxygen from the metal oxide particles. This reduction process creates metal atoms, essentially radicals, that bond to nearby metal particles, creating, upon full reduction of the initial metal oxide particles, metal bridges between metal particles. The process results in the creation of brown, fully de-bonded, solid metal bodies of designed shape and relatively high density (~90%). Simple controls strongly support the RES-SM hypothesis: A solid metal body will form from a compact composed of a mixture of metal and metal oxide particles, but not from a metal particle only compact. 

This is simply an initial study of RES-SM that demonstrates the feasibility of the approach. It leaves many questions. For example: Is there a particle bed thickness at which the process fails to work due to mass transfer limitations of radicals into the bed, or volatile oxides (e.g., CO_2_) exiting the bed? Are there radical producing chemistries which allow treatment at even lower temperatures? Will RES-SM work for all metals and alloys? Can accelerated sintering reproduce results only anticipated for the HIP process? With further development RES-SM could potentially manufacture metal and alloy structures in successive layers in a fashion similar to standard additive manufacturing.

## Figures and Tables

**Figure 1 materials-12-02890-f001:**
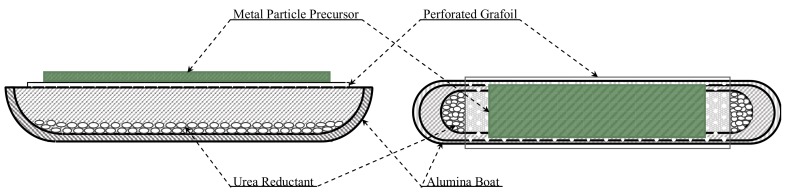
RES-SM Precursor Geometry—As shown, urea is placed in the bottom of an alumina ‘boat’ of dimension 3 cm × 1.5 cm × 1 cm. Grafoil with a high density of pin holes (‘open’ area approximately 25% of total) rests on the top of the alumina boat. The compact, made of Ni and NiO particles, is arranged on top of the Grafoil.

**Figure 2 materials-12-02890-f002:**
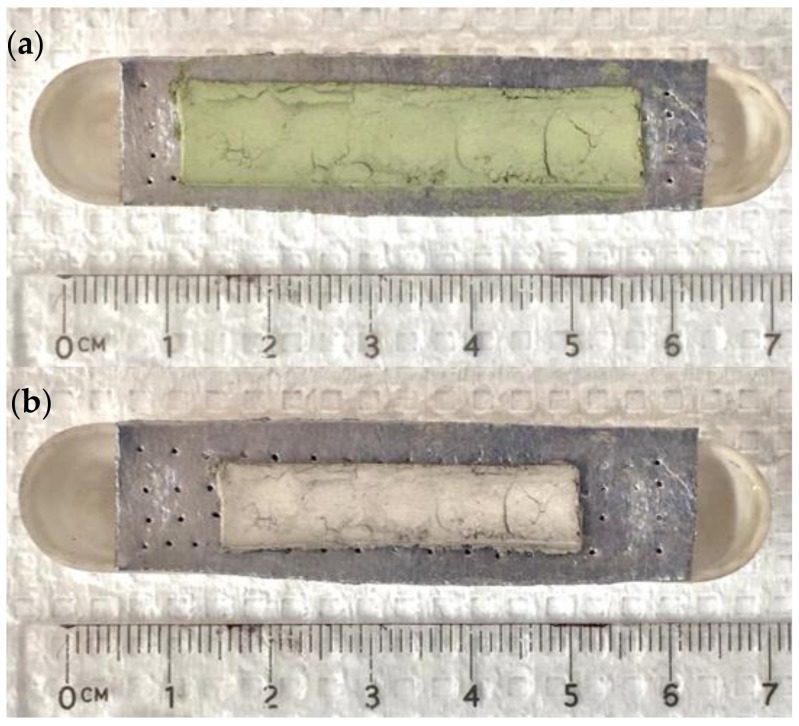
(**a**) On the top is the pre-fired shape created from a 1:1 mix of Ni: NiO. (**b**) On the bottom is the shape after firing. There is clearly shrinkage and reduction of nickel.

**Figure 3 materials-12-02890-f003:**
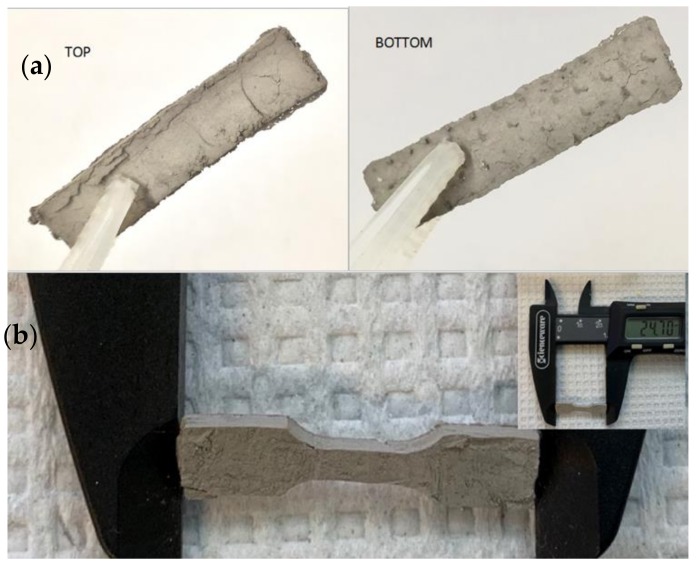
(**a**) The rectangular bodies were shaped by hand with a small spatula. After initial firing they were roughly the same shape as original, but clearly show imperfections. From the TOP it is apparent the body is smooth, with some thin layer formation extending past the main body edges. The BOTTOM view shows ‘dimples’ formed directly above pin holes in the underlying Grafoil. (**b**) Precursor mix (1:1 Ni:NiO) was compressed in a mold created by a 3-D plastics printer. The mold was carefully removed prior to firing. In this example two firings were conducted to ensure full reduction. It is also notable that the final shape (~25 mm long) contains surface imperfections, but the overall shape is faithful to the mold.

**Figure 4 materials-12-02890-f004:**
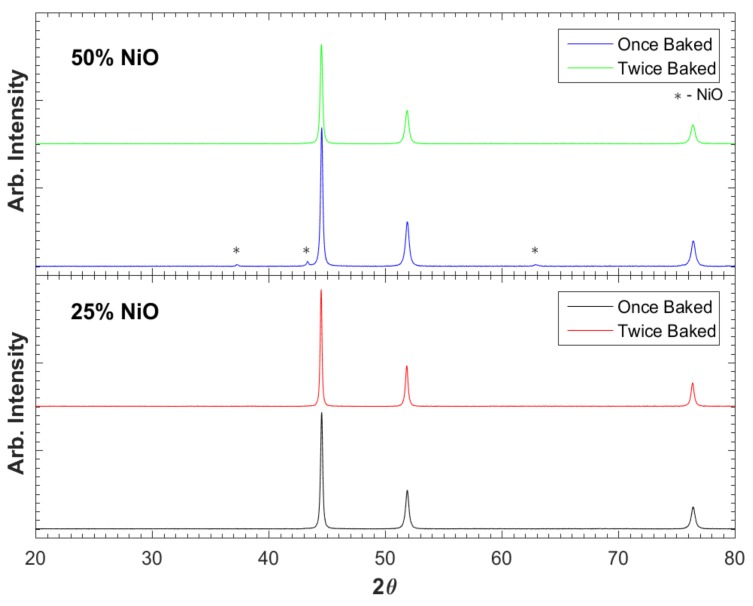
XRD of once baked and twice baked specimens of both 25% NiO and 50% NiO loadings. Unmarked peaks are Ni metal. There is very small oxide component remaining for the 50% NiO sample after the first bake. It is not found in any other case.

**Figure 5 materials-12-02890-f005:**
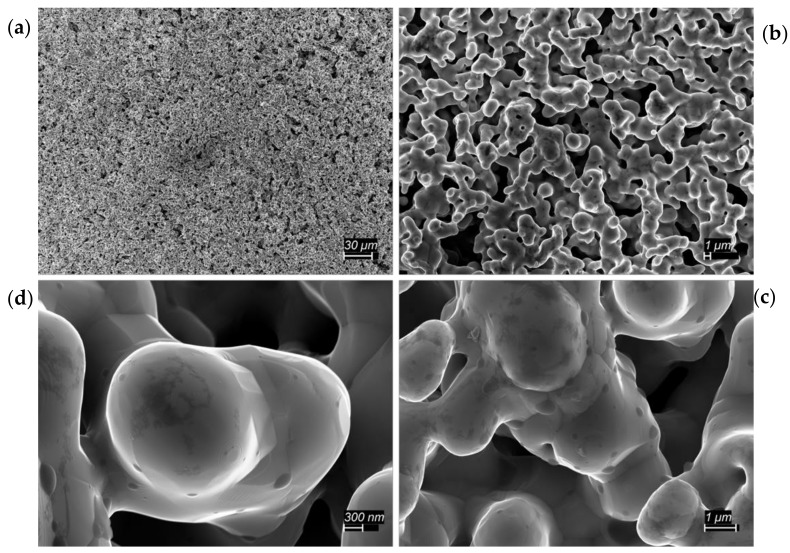
SEM Images of sintered 1:1 Ni:NiO part after 2nd bake. The four figures show the same area with increased magnification. Clearly, all particles are sintered together (**a**) and (**b**). There is no evidence of ‘heterogeneous’ particle composition, and no charging, consistent with conductive metal, observed in the SEM. Virtually all the ‘primary’ particles (**c**) and (**d**) are between 1 and 5 microns.

**Figure 6 materials-12-02890-f006:**
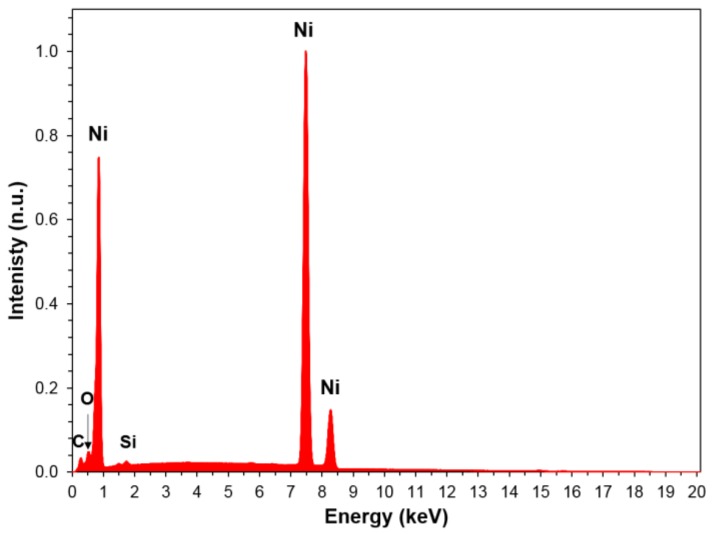
EDS spectra averaged over multiple spots in once-baked 50% Ni:NiO sample.

**Figure 7 materials-12-02890-f007:**
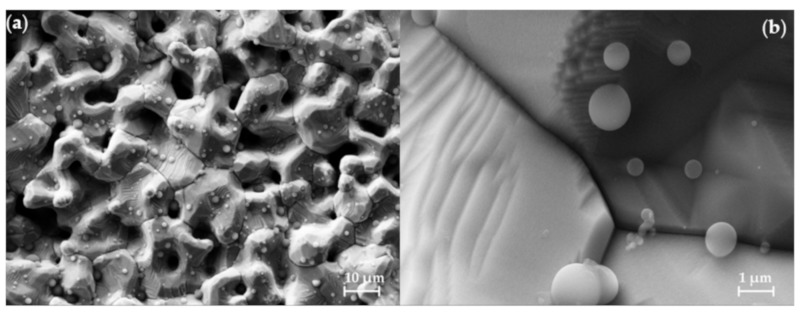
SEM Analysis of a post-HIP sample. After HIP treatment the 50/50 Ni/NiO precursor part was cut and examined in the SEM. It shows no voids equivalent to those observed pre-HIP (Figure 5). (**a**) Low magnification area image and (**b**) higher magnification image showing triple-junction.

**Figure 8 materials-12-02890-f008:**
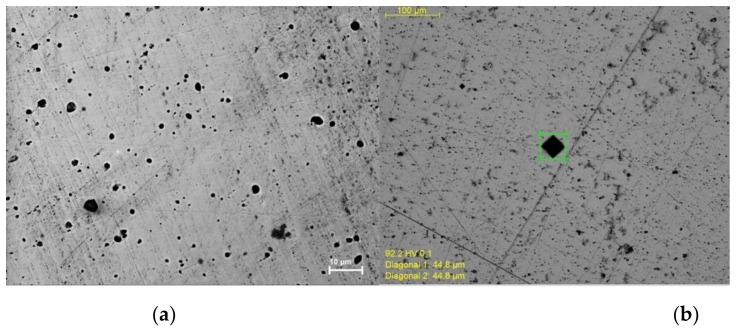
SEM image of polished HIP sample. (**a**) A 50/50 Ni/NiO sample after HIP treatment appears inhomogeneous. There are no longer necks and particles, but rather continuous Ni metal zones and voids. (**b**) Image taken with micro hardness tester showing diamond shaped (~45 µm) hardness indent.

**Table 1 materials-12-02890-t001:** The samples created using the standard protocol.

Composition Ni:NiO wt Ratio (X repeats)	Urea Present	% Metallic Ni ^2^	Self-Supporting Green Body ^2^	% Density After HIP ^4^
1:3 (1×)	Yes	100	Yes	-
1:1 (4×)	Yes	90 ^3^	Yes	94 ^5^
3:1 (3×)	Yes	90 ^3^	Yes	-
9:1 (2×)	Yes	100	Yes	-
1:0	Yes	100	No	-
0:1	Yes	~ 90	No	-
1:1	No	~ 50	No	-
3:1	Yes ^1^	100	No	-

^1^Urea physically mixed with precursor powder. ^2^ After one firing. ^3^ Fully reduced after two firings. ^4^ Density measured by buoyancy and normalized to bulk Ni metal. Not applicable to samples, not HIP treated, with connected pores. ^5^ Based on physical measurements and weight, estimated density normalized to bulk nickel of 1× fired samples was 80+/−5%. See Table 2.

**Table 2 materials-12-02890-t002:** Accelerated Sintering: density and hardness of samples after various treatment regimes.

Sample, 1:1 Ni:NiO Treatment ^1^	Relative Volume, +/−5%	Urea Present	Time at T (min) Beyond 1× Fired	Vickers Hardness
1× Fired	1.00	Yes	0	-
2× Fired	0.90	Yes	20	-
3× Fired	0.87	Yes	40	-
4× Fired	0.82	Yes	60	51.3, 52.7
1× Fired + Heat in Ar	0.90	No	120	25.8, 22.7
1× Fired + HIP	0.80	No	-	86+/−4

^1^ Table shows that firing with standard protocol, urea in alumina boat below sample, leads to faster densification and hardening than simply holding at temperature. Still, the final hardness after 4× firing is far below that achieved using HIP processing.

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
