# Peer review of "Reduction Expansion Synthesis of Sintered Metal"

_materials, 2019, doi:10.3390/ma12182890_

Round 1
Reviewer 1 Report
The paper provides a detailed work on a novel variation of Reduction Expansion Syntesis method to obtain dense metal objects at low temperature and pressure. It is well written and, the design procedures and results are explained in detail. Arising questions in the Results section are answered thoroughly in the Discussion. This paper is ready to be published after two minor corrections:
Line 14: degree sign is missing.
Line 85: ASTM acronym, can you write what it stands for?
Author Response
Response to Reviewer 1:
Thank you. We added the degree sign and gave the official name of the organization that provides standards for materials testing:
ASTM International
Reviewer 2 Report
This paper introduces a new process “reduction expansion synthesis-sintered metal” to produce nickel parts with complex shape, however, the authors failed to present the paper in a scientific way. It is not like a scientific paper in current form. I recommend that the paper should not be accepted for the publication in the present.
Author Response
Response to Reviewer 1:
Note: In responding to Reviewer 2 the authors had to imagine receiving information regarding the basis for rejecting the paper. The Reviewer provided no information…So, this was our best shot at an imagined target.
This work was designed to test a simple hypothesis. The Reduction Expansion Synthesis [1–8]concept can be employed in a novel variation, Reduction Expansion Synthesis-Sintered Metal (RES-SM), to create, quickly, fully reduced ‘brown’ self-supporting, metal only objects of designed shape at ambient pressure, and temperatures hundreds of degrees below the melting temperature of the metal. As described below, this postulate proved true.
The ‘problem under investigation’ is the clearly stated hypothesis. Testing a novel hypothesis (‘novelty of this work’) is at the core of science.
The logic employed in constructing the Introduction: Organize as a science paper. Stop. This is not intended to be a ‘research paper’, nor a business case. As such it begins (see above) with a brief discussion of the general hypothesis, then provides a more detailed version in two stages. The general RES hypothesis is explained:
The general RES concept in broad terms: Chemical radicals are released by thermal decomposition in an inert atmosphere of some simple chemicals such as urea and petroleum gel. The released radicals react with oxygen atoms in nearby solids to form volatile oxygen complexes, leaving reduced metal behind. An exemplary example: A physical mix of urea and metal oxide (e.g. iron) particles, heated to 950 °C in an inert atmosphere creates fully reduced metal particles [2,3]
Next, the specifics of the RES-SM variation to the general RES model, the hypothesis/‘problem’ of the manuscript, is spelled out, with revisions addedfor clarification: :
The production of macroscopic scale, solid metal parts, a simple modification of this general mechanistic hypothesis is advanced: mobile metal atoms will form when metal oxide particles in a compact of metal and metal oxide particles are exposed to a reducing atmosphere. In a process akin to classic Ostwald Ripening [9,10], these atoms, or atomic clusters, effectively radicals, will migrate to, then bond to, existing metal surfaces in the immediate vicinity to create larger metal bodies. A similar ‘metal radical’ model is used to explain the growth of large metal particles in the gas phase during catalytic etching [11], as well as enhanced rates of particle growth in supported catalysts under reaction conditions [12]. A test of this hypothesis: Using the RES-SM method a solid object can be generated from a mixture of metal oxide and metal particles , and not from metal particles alone.
.
Next, a brief summary of observations consistent with expectations of the hypothesis is provided:
Observations in this study regarding brown body formation are consistent with the above model. Most important, larger bodies do form from the original metal/metal oxide compact with a morphology consisting of the original metal particles joined by metal ‘necks’. That is, the ‘brown body’ consists solely of a network of metal particles linked by ‘necks’, and a homogenous distribution of void spaces of about the same size as the original particles. Moreover; the final metallic object has the same ‘shape’ as the original compact but is smaller in size. The compact shrinks in a uniform fashion such that relative dimensions remain nearly unchanged. Hot isostatic pressing creates a part with hardness equal that of metal made by standard casting. In contrast, using the RES-SM protocol a solid object does not form from a metal particles only compact.
The sequence of topics is designed to minimize drama. There are no secrets to be revealed later. The Method: Provide a succinct hypothesis up front such that the narrative organization, designed to test the hypothesis, is clear. Given this introductory material the reader is prepared to understand/critique each data point/concept, etc. as the reader progresses through the Results and Discussion sections. Is the data consistent with the Hypothesis? This well worn organization approach is employed: ‘Tell them what you are going to tell them, tell them, and finally tell them what you told them’.
As this is a science paper, testing a clearly novel hypothesis, comparisons of ‘techniques’ are not necessary. Still, the Discussion does contain many comparison with the closest technological approach. In fact, the comparisons with other techniques represent ~10% of the narrative words:
From Section 4.1:
It is also important to note that the process included some fundamentally novel features. First, a ‘brown’ [21]body is formed from a mix of metal and metal oxide. In all other systems, including particle injection molding (PIM) and all M-AM technologies, the metal precursors are entirely reduced metal particles. Second, no ‘binder’ is required as per PIM or metal containing paste based AM techniques. This is particularly important as ‘debonding’, that is removal of the binder, generally an organic material, is the slowest, most complex step in these PIM processes. Thus, in the terminology of PIM, the body formed in RES-SM is a ‘brown’ body, that is it is equivalent to a de-bonded PIM generated body, a body consisting only of metal. Third, unlike M-AM, at no point is there a need to reach the metal melting point. Indeed, in this study, a solid nickel body formed at 900 °C, whereas the nickel melting temperature is 1455 °C.
From Section 4.4:
At the temperature employed herein, 900 °C for RES-SM firing, nickel particles (no oxide particles)were shown in prior studies to sinter without a reducing agent at high pressure [22,23]. It is notable that quantitative comparison with earlier studies is not possible because all of them were conducted at pressures at least 1000 X higher than that employed herein. Moreover, it is clear that high pressures (ca. >1000 atm), even at 600 °C, dramatically impacts densification [22].
From Section 4.5
4.5 Additive Manufacturing
RES-SM can be considered a form of metal additive manufacturing as no material is ever removed. Notably, it is related to another ‘hybrid AM’ technology based on creating molds with P-AM and adding metal. Specifically, parts are created by pouring liquid metal from low temperature melting metals (bismuth, lead, tin, cadmium, and indium) into plastic molds with a top operative temperature of ~180 °C [24]. The advantages of RES-SM are: i) low cost of the hardware, ii) lower cost of the precursors (metal oxides are less expensive than metals), iii) simplicity, iv) scalability (e.g. many parts can be manufactured simultaneously in a single furnace), v) rapidity as only 20 minutes are required for processing a part of any size, and v) the wide pallet of metals (e.g. Ni, Fe, Co, Sn, Cr) and alloys that likely can be used to create solid objects with the technique.
Also note added to the end of Section 4.5
Given RES-SM is a form of M-AM, could it potentially be developed into a competitive technology? At this stage of RES-SM development an answer is premature. Still it is reasonable to point out that that relative to one technique, M-AM by selective laser raster of a particle bed, first introduced thirty years ago, there may be advantages to RES-SM. To wit: Assuming equal mechanical properties can be achieved using either RES-SM, or commercial layer-by-layer laser/e-beam sintering, at present the former represents a far smaller (ca. 5%) capital investment, is far faster (~20 min., any size object), and potentially can be used to make large objects. The latter, presently, has the advantages of more optimized operation, better precision, hence more detail in the product, clearly can make objects many centimeters in all dimensions,and likely creates a superior product. Also, any M-AM process that requires high pressure and/or debonding (e.g. PIM), is likely slower and more costly than a further developed RES-SM. However, it is clear more M-AM technologies will be introduced in the near future, that all M-AM technologies are rapidly transforming/improving, so all comparisons and projections are fraught with uncertainty.
Comparison to ‘sintering’: As we note, sintering temperature is related to pressure. There is no literature regarding sintering at ambient pressure. All data in the literature is recorded during sintering at pressures at least 1000 X higher than those employed herein. As noted, we only found small ‘chunks’ of sintered metal from pure Ni metal particle beds after prolonged sintering at 950 C.
“At the temperature employed herein, 900 °C for RES-SM firing, compacts of pure nickel particles(no oxide particles), were shown in prior studies to sinter without a reducing agent at high pressure [22,23]. It is notable that quantitative comparison with earlier studies is not possible because all of them were conducted at pressures at least 1000 X higher than that employed herein. Moreover, it is clear that high pressures (ca. >1000 atm), even at 600 °C, dramatically impacts densification [22]. Still this raises a question: After the first firing, necessary to reduce all the nickel oxide, is there any advantage to subsequent firing? Table II clearly suggests there is a case to be made for accelerated sintering using standard RES-SM firing. Indeed, the hardness and ‘shrinkage’ after 60 minutes at temperature in the RES-SM process is far better than that achieved in inert gas at the same temperature for 120 minutes. Possibly further study will reveal a path to full densification, without HIP, at ‘low’ temperature and ambient pressure.”
.
We re-worked the conclusions to make the overall construction of the manuscript ‘fully parallel’:
The data strongly support the hypothesis of RES-SM: A mixture of metal and metal oxide particles, exposed to the volatile species created by the thermal decomposition of urea, remove oxygen from the metal oxide particles. This reduction process creates metal atoms, essentially radicals, that bond to nearby metal particles, creating, upon full reduction of the initial metal oxide particles, metal bridges between metal particles. The process results in the creation of brown, fully de-bonded, solid metal bodies of designed shape and relatively high density (~90%). Simple controls strongly support the RES-SM hypothesis: A solid metal body will form from a compact composed of a mixture of metal and metal oxide particles, but not from a metal particle only compact.
Style suggested is personal to the reviewer. My experience as the author of more than 160 reviewed publications and 36 issued patents, reviewer of many dozens of theses, and hundreds of manuscripts, is that there are a range of possible organization strategies. In my review work I refrain from imposing personal preferences.
Reviewer 3 Report
The paper entitled "Reduction Expansion Synthesis of Sintered Metal” by Daniels et al. deals with a new sintering technique and the production of a nickel part using it. The technique is analysed and nickel parts are characterized (composition, microhardness, etc.). Results are in the scope of the Materials journal. These are interesting, but after reading the paper, I have some comments about it:
GENERAL COMMENTS:
1) The introduction should be rewritten. First, the introduction should summarize the state of the subject under investigation (in this case, sintering for production of object). This section is not a summary of the results which are going to be described in the following pages. It is an introduction to determine which is the problem under investigation and which is the state of the art in this regard. Second, the introduction should review the main works on this topic to highlight the novelty of this work. In the current state, this section is not a suitable for a research paper.
2) Discussion section should include a comparison with other techniques. In this work only the advantages of this technique are presented, but not the disadvantages (notice, that many cracks are formed after firing?) or a proper comparison with the current sintering techniques. As this can be considered an additive manufacturing technique, the mechanical properties (e.g. Young’s modulus, ultimate tensile strength, etc. ) of the final objects should be measured and compared with that of the bulk material.
3) The conclusions section should also be rewritten. This section is a summary of the main findings of the work, not a section discussing the questions to be addressed in future works. These questions should be discussed in the discussion section.
4) Finally, it is not clear from this work if this technique is also valid for other materials (which ones?) and if the manufacturing procedure would be similar or what changes should be made.
PARTICULAR COMMENTS
1) (Page 2) Please, add the purity of the chemicals (e.g. for Hydrogen Peroxide, urea) used in this study.
2) (Page 2) Please, add a table with the weight ratios of the metal/metal oxide tested in this study.
3) (Page 3) Please, add the heating rate (heating up to 950 °C, but at which rate?) and cooling rates (cool at ambient temperature from 950ºC, but at which rate?)
4) (Page 6) Please, determine the different phases from the XRD analyses.
5) (Page 7) EDS analyses are presented in Fig. 6. Was the EDS performed in a semi-quantitative or in a quantitative mode (i.e. comparison with standards)?
6) (Page 9) In the text, it is asserted that “a solid nickel body formed at 900 °C, whereas the nickel melting temperature is 1455 °C”. Notice that for sintering, it is not required to reach the melting temperature.
Author Response
Response to Reviewer 3:
Reviewer Comment 1:First, the introduction should summarize the state of the subject under investigation (in this case, sintering for production of object).
Response 1: The subject of this paper is not primarily ‘sintering for the production of object’. The subject is a novel hypothesis, which is correct for a scientific paper. Although the range of activities considered ’science’ is broad, first among all definitions of a scientific activity, is to propose a hypothesis, and test it. Clearly, this is the approach taken throughout, including the first paragraph:
This work was designed to test a simple hypothesis. The Reduction Expansion Synthesis [1–8]concept can be employed in a novel variation, Reduction Expansion Synthesis-Sintered Metal (RES-SM), to create, quickly, fully reduced ‘brown’ self-supporting, metal only objects of designed shape at ambient pressure, and temperatures hundreds of degrees below the melting temperature of the metal. As described below, this postulate proved true.
Reviewer Comment 2: This section is not a summary of the results which are going to be described in the following pages. It is an introduction to determine which is the problem under investigation and which is the state of the art in this regard. Second, the introduction should review the main works on this topic to highlight the novelty of this work. In the current state, this section is not a suitable for a research paper.
The ‘problem under investigation’ is the clearly stated hypothesis. Testing a novel hypothesis (‘novelty of this work’) is at the core of science.
The logic employed in constructing the Introduction: Organize as a science paper. Stop. This is not intended to be a ‘research paper’, nor a business case. As such it begins (see above) with a brief discussion of the general hypothesis, then provides a more detailed version in two stages. The general RES hypothesis is explained:
The general RES concept in broad terms: Chemical radicals are released by thermal decomposition in an inert atmosphere of some simple chemicals such as urea and petroleum gel. The released radicals react with oxygen atoms in nearby solids to form volatile oxygen complexes, leaving reduced metal behind. An exemplary example: A physical mix of urea and metal oxide (e.g. iron) particles, heated to 950 °C in an inert atmosphere creates fully reduced metal particles [2,3]
Next, the specifics of the RES-SM variation to the general RES model, the hypothesis/‘problem’ of the manuscript, is spelled out, with revisions addedfor clarification: :
The production of macroscopic scale, solid metal parts, a simple modification of this general mechanistic hypothesis is advanced: mobile metal atoms will form when metal oxide particles in a compact of metal and metal oxide particles are exposed to a reducing atmosphere. In a process akin to classic Ostwald Ripening [9,10], these atoms, or atomic clusters, effectively radicals, will migrate to, then bond to, existing metal surfaces in the immediate vicinity to create larger metal bodies. A similar ‘metal radical’ model is used to explain the growth of large metal particles in the gas phase during catalytic etching [11], as well as enhanced rates of particle growth in supported catalysts under reaction conditions [12]. A test of this hypothesis: Using the RES-SM method a solid object can be generated from a mixture of metal oxide and metal particles , and not from metal particles alone.
.
Next, a brief summary of observations consistent with expectations of the hypothesis is provided:
Observations in this study regarding brown body formation are consistent with the above model. Most important, larger bodies do form from the original metal/metal oxide compact with a morphology consisting of the original metal particles joined by metal ‘necks’. That is, the ‘brown body’ consists solely of a network of metal particles linked by ‘necks’, and a homogenous distribution of void spaces of about the same size as the original particles. Moreover; the final metallic object has the same ‘shape’ as the original compact but is smaller in size. The compact shrinks in a uniform fashion such that relative dimensions remain nearly unchanged. Hot isostatic pressing creates a part with hardness equal that of metal made by standard casting. In contrast, using the RES-SM protocol a solid object does not form from a metal particles only compact.
The sequence of topics is designed to minimize drama. There are no secrets to be revealed later. The Method: Provide a succinct hypothesis up front such that the narrative organization, designed to test the hypothesis, is clear. Given this introductory material the reader is prepared to understand/critique each data point/concept, etc. as the reader progresses through the Results and Discussion sections. Is the data consistent with the Hypothesis? This well worn organization approach is employed: ‘Tell them what you are going to tell them, tell them, and finally tell them what you told them’.
Reviwer Comment 3: 2) Discussion section should include a comparison with other techniques.
Response: As this is a science paper, testing a clearly novel hypothesis, comparisons of ‘techniques’ are not necessary. Still, the Discussion does contain many comparison with the closest technological approach. In fact, the comparisons with other techniques represent ~10% of the narrative words:
From Section 4.1:
It is also important to note that the process included some fundamentally novel features. First, a ‘brown’ [21]body is formed from a mix of metal and metal oxide. In all other systems, including particle injection molding (PIM) and all M-AM technologies, the metal precursors are entirely reduced metal particles. Second, no ‘binder’ is required as per PIM or metal containing paste based AM techniques. This is particularly important as ‘debonding’, that is removal of the binder, generally an organic material, is the slowest, most complex step in these PIM processes. Thus, in the terminology of PIM, the body formed in RES-SM is a ‘brown’ body, that is it is equivalent to a de-bonded PIM generated body, a body consisting only of metal. Third, unlike M-AM, at no point is there a need to reach the metal melting point. Indeed, in this study, a solid nickel body formed at 900 °C, whereas the nickel melting temperature is 1455 °C.
From Section 4.4:
At the temperature employed herein, 900 °C for RES-SM firing, nickel particles (no oxide particles)were shown in prior studies to sinter without a reducing agent at high pressure [22,23]. It is notable that quantitative comparison with earlier studies is not possible because all of them were conducted at pressures at least 1000 X higher than that employed herein. Moreover, it is clear that high pressures (ca. >1000 atm), even at 600 °C, dramatically impacts densification [22].
From Section 4.5
4.5 Additive Manufacturing
RES-SM can be considered a form of metal additive manufacturing as no material is ever removed. Notably, it is related to another ‘hybrid AM’ technology based on creating molds with P-AM and adding metal. Specifically, parts are created by pouring liquid metal from low temperature melting metals (bismuth, lead, tin, cadmium, and indium) into plastic molds with a top operative temperature of ~180 °C [24]. The advantages of RES-SM are: i) low cost of the hardware, ii) lower cost of the precursors (metal oxides are less expensive than metals), iii) simplicity, iv) scalability (e.g. many parts can be manufactured simultaneously in a single furnace), v) rapidity as only 20 minutes are required for processing a part of any size, and v) the wide pallet of metals (e.g. Ni, Fe, Co, Sn, Cr) and alloys that likely can be used to create solid objects with the technique.
Also note added to the end of Section 4.5 as a response to the reviewer’s comment:
Given RES-SM is a form of M-AM, could it potentially be developed into a competitive technology? At this stage of RES-SM development an answer is premature. Still it is reasonable to point out that that relative to one technique, M-AM by selective laser raster of a particle bed, first introduced thirty years ago, there may be advantages to RES-SM. To wit: Assuming equal mechanical properties can be achieved using either RES-SM, or commercial layer-by-layer laser/e-beam sintering, at present the former represents a far smaller (ca. 5%) capital investment, is far faster (~20 min., any size object), and potentially can be used to make large objects. The latter, presently, has the advantages of more optimized operation, better precision, hence more detail in the product, clearly can make objects many centimeters in all dimensions,and likely creates a superior product. Also, any M-AM process that requires high pressure and/or debonding (e.g. PIM), is likely slower and more costly than a further developed RES-SM. However, it is clear more M-AM technologies will be introduced in the near future, that all M-AM technologies are rapidly transforming/improving, so all comparisons and projections are fraught with uncertainty.
3) In this work only the advantages of this technique are presented, but not the disadvantages (notice, that many cracks are formed after firing?) or a proper comparison with the current sintering techniques.
Comparison to ‘sintering’: As we note, sintering temperature is related to pressure. There is no literature regarding sintering at ambient pressure. All data in the literature is recorded during sintering at pressures at least 1000 X higher than those employed herein. As noted, we only found small ‘chunks’ of sintered metal from pure Ni metal particle beds after prolonged sintering at 950 C.
“At the temperature employed herein, 900 °C for RES-SM firing, compacts of pure nickel particles(no oxide particles), were shown in prior studies to sinter without a reducing agent at high pressure [22,23]. It is notable that quantitative comparison with earlier studies is not possible because all of them were conducted at pressures at least 1000 X higher than that employed herein. Moreover, it is clear that high pressures (ca. >1000 atm), even at 600 °C, dramatically impacts densification [22]. Still this raises a question: After the first firing, necessary to reduce all the nickel oxide, is there any advantage to subsequent firing? Table II clearly suggests there is a case to be made for accelerated sintering using standard RES-SM firing. Indeed, the hardness and ‘shrinkage’ after 60 minutes at temperature in the RES-SM process is far better than that achieved in inert gas at the same temperature for 120 minutes. Possibly further study will reveal a path to full densification, without HIP, at ‘low’ temperature and ambient pressure.”
.
4) As this can be considered an additive manufacturing technique, the mechanical properties (e.g. Young’s modulus, ultimate tensile strength, etc. ) of the final objects should be measured and compared with that of the bulk material.
We employed hardness testing, a means frequently employed, to test the sintered objects. See Table II and Fig. 8. And yes, we hoped to make measurements with an Instron, but the instrument broke and replacement cost for one of the proper size is >$10K. Later.
5) The conclusions section should also be rewritten. This section is a summary of the main findings of the work, not a section discussing the questions to be addressed in future works. These questions should be discussed in the discussion section.
We re-worked the conclusions to make the overall construction of the manuscript ‘fully parallel’:
The data strongly support the hypothesis of RES-SM: A mixture of metal and metal oxide particles, exposed to the volatile species created by the thermal decomposition of urea, remove oxygen from the metal oxide particles. This reduction process creates metal atoms, essentially radicals, that bond to nearby metal particles, creating, upon full reduction of the initial metal oxide particles, metal bridges between metal particles. The process results in the creation of brown, fully de-bonded, solid metal bodies of designed shape and relatively high density (~90%). Simple controls strongly support the RES-SM hypothesis: A solid metal body will form from a compact composed of a mixture of metal and metal oxide particles, but not from a metal particle only compact.
Style suggested is personal to the reviewer. My experience as the author of more than 160 reviewed publications and 36 issued patents, reviewer of many dozens of theses, and hundreds of manuscripts, is that there are a range of possible organization strategies. In my review work I refrain from imposing personal preferences.
6) Finally, it is not clear from this work if this technique is also valid for other materials (which ones?) andif the manufacturing procedure would be similar or what changes should be made.
Exactly. This is a first study. We invite the reviewer to participate in further study of this process.
PARTICULAR COMMENTS
(Page 2) Please, add the purity of the chemicals (e.g. for Hydrogen Peroxide, urea) used in this study.
30% in water
2) (Page 2) Please, add a table with the weight ratios of the metal/metal oxide tested in this study.
See Table I, first column
3) (Page 3) Please, add the heating rate (heating up to 950 °C, but at which rate?) and cooling rates (cool at ambient temperature from 950ºC, but at which rate?)
4) (Page 6) Please, determine the different phases from the XRD analyses.
Done.
5) (Page 7) EDS analyses are presented in Fig. 6. Was the EDS performed in a semi-quantitative or in a quantitative mode (i.e. comparison with standards)?
Semi-quantitative, but based on a great deal of experience and obviousness…
6) (Page 9) In the text, it is asserted that “a solid nickel body formed at 900 °C, whereas the nickel melting temperature is 1455 °C”. Notice that for sintering, it is not required to reach the melting temperature.
Addressed in manuscript: At the temperature employed herein, 900 °C for RES-SM firing, nickel particles (no oxide particles)were shown in prior studies to sinter without a reducing agent at high pressure [22,23]. It is notable that quantitative comparison with earlier studies is not possible because all of them were conducted at pressures at least 1000 X higher than that employed herein. Moreover, it is clear that high pressures (ca. >1000 atm), even at 600 °C, dramatically impacts densification [22].
Round 2
Reviewer 3 Report
In its present state, the paper entitled “Reduction Expansion Synthesis of Sintered Metal” is acceptable for publication. Authors have addressed most of the concerns commented by the reviewer. However, as pointed out in the first review report, I would modify the introduction to be more informative; for example, the potential application of this technique in additive manufacturing it is evident. Then, I could be added a brief discussion on this topic in the introduction, and I would comment how this technique can address some problems of other additive manufacturing techniques.
Finally, prior to the publication of this paper, please, correct the References section: Ref. [6] has not all the publication details (publication year, editorial, etc.)